

# RNA sequencing of CD4 T-cells reveals the relationships between lncRNA-mRNA co-expression in elite controller vs. HIV-positive infected patients

Chaoyu Chen, Xiangyun Lu and Nanping Wu

State Key Laboratory for Diagnosis and Treatment of Infectious Diseases, National Clinical Research Center for Infectious Diseases, Collaborative Innovation Center for Diagnosis and Treatment of Infectious Diseases, The First Affiliated Hospital, College of Medicine, Zhejiang University, Hangzhou, Zhejiang, China

## ABSTRACT

**Background**. Elite controller refers to a patient with human immunodeficiency virus infection with an undetected viral load in the absence of highly active antiretroviral therapy. Studies on gene expression and regulation in these individuals are limited but significant, and have helped researchers and clinicians to understand the interrelationships between HIV and its host.

**Methods**. We collected CD4 T-cell samples from two elite controllers (ECs), two HIV-positive infected patients (HPs), and two healthy controls (HCs) to perform second-generation transcriptome sequencing. Using the Cufflinks software, we calculated the Fragments Per Kilobase of transcript per Million fragments mapped (FPKM) and identified differentially expressed (DE) mRNAs and long non-coding RNAs (lncRNAs), with corrected $P$ value $< 0.05$ (based on a false discovery rate (FDR) $< 0.05$). We then constructed a protein-protein interaction network using cytoHubba and a long non-coding RNA-mRNA co-expression network based on the Pearson correlation coefficient.

**Results**. In total, 1109 linear correlations of DE lncRNAs targeting DE mRNAs were found and several interesting interactions were identified as being associated with viral infections and immune responses within the networks based on these correlations. Among these lncRNA-mRNA relationships, hub mRNAs including *HDAC6*, *MAPK8*, *MAPK9*, *ATM* and their corresponding annotated co-expressed lncRNAs presented strong correlations with the MAPK-NF-kappa B pathway, which plays a role in the reactivation and replication of the virus.

**Conclusions**. Using RNA-sequencing, we systematically analyzed the expression profiles of lncRNAs and mRNAs from CD4+ T cells from ECs, HPs, and HCs, and constructed a co-expression network based on the relationships among DE transcripts and database annotations. This was the first study to examine gene transcription in elite controllers and to study their functional relationships. Our results provide a reference for subsequent functional verification at the molecular or cellular level.

Corresponding author
Nanping Wu, flwnp@zju.edu.cn

## INTRODUCTION

Human immunodeficiency virus (HIV) has been studied for more than 30 years, and research has found that HIV invades host CD4+ cells, integrates its own DNA into the host genome, and establishes a reservoir in the early stage, followed by massive replication, which destroys the normal immune system function of the host, and triggers various concurrent symptoms (*Sabin & Lundgren, 2013*; *Eisele & Siliciano, 2012*). Ultimately, HIV infection can cause the host to die. Although highly active antiretroviral theraphy (HAART) can almost completely reconstitute the immune function of the host, it cannot eradicate the latent HIV pool in the host, which always maintains a low level of virus replication (*Murray et al., 2016*). However, among HIV-infected individuals, a small number of patients are found to be inherently resistant to viral replication, and spontaneously induce suppression of the latent pools without HAART, resulting in an undetectable viral load in plasma for a long period (*Migueles & Connors, 2010*; *Gonzalo-Gil, Ikediobi & Sutton, 2017*). These patients, termed elite controllers (ECs), have attracted significant research interest. Studying the characteristics of the autoimmune factors of ECs is anticipated to identify important factors that control virus replication. Such valuable information could lead to novel methods to treat and alleviate HIV infection.

Long-noncoding RNAs (lncRNAs) are transcribed RNA molecules greater than 200 nt in length that regulate gene expression by diverse, but as yet not completely understood, mechanisms (*Wilusz, Sunwoo & Spector, 2009*). Although the function of most lncRNAs is unknown, several have been shown to regulate gene expression at multiple levels, from DNA to phenotype (*Wapinski & Chang, 2011*). Through a variety of means, including *cis* (near the site of lncRNA production) or *trans* (co-expressed with their target gene) mechanisms, lncRNAs play a vital role in many biological processes (*Wang & Chang, 2011*). Studies on lncRNAs have become a hotspot in current non-coding RNA research.

Recent studies focused on the role of lncRNAs in HIV pathogenesis, especially the relationship between lncRNA regulation of gene expression and viral infection, replication, and latency (*Trypsteen et al., 2016*; *Lazar, Morris & Saayman, 2015*; *Saayman et al., 2014*; *Nair, Sagar & Pilakka-Kanthikeel, 2016*). A few lncRNAs have been characterized and proven to be closely associated with HIV, for example, the knockdown of host lncRNA *NEAT1* enhanced virus production by increasing nucleus-to-cytoplasm export of Rev-dependent instability element (INS)-containing HIV-1 mRNAs (*Zhang et al., 2013*). The knockdown of lncRNA *NRON* enhanced HIV-1 replication through increased activity of nuclear factor of activated T-cells (NFAT) and the viral long terminal repeat (LTR) (*Imam et al., 2015*). Similarly, lncRNA *MALAT1* releases epigenetic silencing of HIV-1 replication by displacing the polycomb repressive complex 2 from binding to the LTR promoter (*Qu et al., 2019*). Furthermore, an HIV-encoded antisense lncRNA, *ASP-L*, was proven to promote latency HIV (*Saayman et al., 2014*; *Kobayashi-Ishihara et al., 2012*). In addition, *HEAL* is a broad enhancer of multiple HIV-1 strains because depletion of *HEAL* inhibited X4, R5, and dual-tropic HIV replication, which was rescued by HEAL overexpression (*Ti-Chun et al., 2019*). Although the mechanism of lncRNA function is sometimes elusory and unpredictable, many current studies, which are limited to the

screening and functional prediction stage based on chip or sequencing results, also provide us with valuable information and from the basis for mechanistic research.

In the present study, we conducted a transcriptomics investigation for HIV ECs, to identify differences in the transcriptional expression profiles between ECs and HIV-positive infected patients as compared with healthy controls. The obtained lncRNA-mRNA co-expression network revealed the possible role of the functional relationships between lncRNAs and mRNAs in the ability of ECs to inhibit viral replication.

## METHODS

### Ethics statement
Ethical approval was granted by the Ethics Committee of the First Affiliated Hospital, College of medicine, Zhejiang University. The Reference number is 2017-338. All subjects provided written, informed consent. In addition, all participating medical institutions provided local institutional review board approval.

### Cohort and patients
In total, we recruited 196 patients into our cohort, from different provinces of China, who were diagnosed in Qingchun Hospital of Zhejiang Province from January 2000 to January 2018, and received, or volunteered to rejected HAART. The viral loads of the patients were detected using the Cobas Taqman system (Roche, Basle, Switzerland). The healthy controls (HCs) were randomly selected from hospital clinics and were not suffering from any diseases. We chose two ECs, two HIV-positive infected patients (HPs), and two HCs of similar ages, sex, and traditional risk factors to perform second generation transcriptome sequencing. ECs were defined as HIV-infected patients with an undetectable viral load in the absence of HAART for nearly 10 years.

### Sample collection and preparation
Peripheral blood mononuclear cells (PBMCs) were isolated from 3–5 mL EDTA-K2+ anticoagulant venous blood using density gradient centrifugation. Primary CD4+ T cells were purified through negative selection using a CD4+ T cell isolation Kit (Miltenyi Biotec, Bergisch, Gladbach, Germany). According to flow cytometry analysis, the purity of the separated CD4+ T cells was more than 90%. Total RNA was extracted using the TRIzol reagent (Life Technologies, Carlsbad, CA, USA). RNA purity was checked using a NanoPhotometer® spectrophotometer (IMPLEN, München, Germany). The RNA concentration was measured using a Qubit® RNA Assay Kit in Qubit® 2.0 Fluorometer (Life Technologies) and its integrity was assessed using an RNA Nano 6000 Assay Kit for the Bioanalyzer 2100 system (Agilent Technologies, Santa Clara, CA, USA).

### RNA library construction and RNA sequencing
Ribosomal RNA was removed using a Epicentre Ribo-zero™ rRNA Removal Kit (Epicentre, Madison, WI, USA), and the rRNA free residue was cleaned using ethanol precipitation. Subsequently, sequencing libraries were generated using the rRNA depleted RNA and an NEBNext® Ultra™ Directional RNA Library Prep Kit for Illumina® (NEB,

Ipswich, MA, USA) following manufacturer's recommendations. After cDNA synthesis and purification, clustering of the index-coded samples was performed on a cBot Cluster Generation System using a TruSeq PE Cluster Kit v3-cBot-HS (Illumina, San Diego, CA, USA) according to the manufacturer's instructions. The libraries were then sequenced on an Illumina Hiseq 4000 platform and raw data were generated.

## RNA sequencing analysis

Clean data were obtained by removing reads containing adapters, reads only containing poly-N, and low quality reads from the raw data. After quality control, paired-end clean reads were aligned to the reference genome downloaded from website using HISAT2 (v2.0.4) (*Pertea et al., 2016*). Cufflinks (v2.1.1) was used to calculate the FPKM (expected number of Fragments Per Kilobase of transcript sequence per Millions base pairs sequenced) values of both lncRNAs and mRNAs in each sample (*Trapnell et al., 2010*). FPKM is the number of fragments from a gene per thousand bases in each million fragments. It takes into account the sequencing depth and the effect of gene length on the fragment count (Figs. S3, S4). Transcripts with FPKM values below 1.0 in any one of the samples were excluded. A summary of gene FPKM values were obtained by combining the FPKM values of transcripts. Cuffdiff (v2.1.1) provided statistical routines to determine differential expression in the digital transcript or gene expression data using a model based on the negative binomial distribution. Transcripts with a corrected $P$ value <0.05 (false discovery rate (FDR) <0.05) were assigned as differentially expressed. We plotted a heatmap to observe the clustering between the samples and the genes using Multi Experiment Viewer (v4.9.0). All raw data was uploaded in the *.bam* format and stored in the SRA database at the NCBI, with the accession number PRJNA575767.

## LncRNA-mRNA co-expression network analysis

A *Trans* role is defined as an lncRNA binding to a target DNA as an RNA:DNA heteroduplex, as an RNA:DNA:DNA triplex, or RNA recognition of specific chromatin-like complex surfaces (*Hung & Chang, 2010*). We calculated the expression correlation between lncRNAs and mRNAs using the Pearson correlation test for target gene prediction and the results were expressed as Pearson correlation coefficients. To narrow the scope of investigation and focus on the most important transcripts, we first assessed the interaction among all differentially expressed (DE) mRNAs using the STRING database (v11.0) and constructed a protein-protein interaction (PPI) network using the cytoHubba package in Cytoscape (v3.7.1). After considering 12 synthetic algorithms, the intersection of the top 50 genes yielded 24 hub genes, which occupied a central position in the network of all DE mRNAs. After identifying and retaining the DE mRNAs and lncRNAs in all lncRNA-mRNA relations, we screened 1109 pairs of exact relationships between the two groups and constructed a co-expression network based on the Pearson relation coefficient (all >0.95, $p < 0.001$). Then, we identified the lncRNAs and their associated mRNAs for further functional enrichment analysis. The PPI and lncRNA-mRNA network diagrams were drawn using the Cytoscape software (v3.7.1).

## Gene ontology and KEGG pathways analysis

By jointly using the David database (v6.8) and KOBAS database (v3.0), gene ontology (GO) enrichment and Kyoto Encyclopedia of Genes and Genomes (KEGG) pathways analysis of all DE mRNAs and DE lncRNA-co-expressed DE mRNAs were implemented to interpret the biological meaning of the transcripts. GO terms and KEGG pathways with corrected $P$ values less than 0.05 (FDR < 0.05) were considered significantly enriched by the differentially expressed RNAs.

## Quantitative real time PCR validation

We used quantitative real-time PCR (qRT-PCR) with SYBR green analysis to validate the accuracy of the sequencing results. To validate random DE mRNAs, total RNA was extracted using TRIzol reagent (Life Technologies). QRT-PCR was performed with the iQ$^{TM}$ SYBR® Green Supermix (Bio-Rad, Hercules, CA, USA). *ACTB* (encoding beta-actin) was used as the endogenous control for the mRNA analysis. The expression levels of mRNA were calculated based on the change in cycling threshold using the of $2^{-\Delta Ct}$ method (*Pertea et al., 2016*). GraphPad Prism 7 (GraphPad Software, Inc., La Jolla, CA, USA) was used to perform the qRT-PCR statistical analysis. The non-parametric Mann–Whitney U test was used to compare between group distributions.

# RESULTS

## Subjects

A total of 196 individuals infected with HIV were enrolled in our cohort, including patients under treatment (181, 92.4%), HIV-positive infected patients (13, 6.6%) and elite controllers (2, 1.0%). In addition, we recruited two healthy individuals from the out-patient department of the hospital. Most patients were infected with the virus through heterosexual contact, followed by intravenous infection, and homosexual transmission. Based on the basic principle of intra-group identity, we determined three men and three women who were assigned to each of the three study groups as study subjects. Their ages ranged from 39 to 54 and were free of other diseases, such as tuberculosis, diabetes, or hepatitis. Specific details of the clinical characteristics of the participants are shown in Table S1. Descriptive statistics are reported as counts (percentage) for dichotomous and categorical variables, and the median (the 25th percentile and 75th percentile) for continuous variables. The experimental design and analysis process of this research are shown in a flowchart (Fig. 1).

## Identification of DE transcripts between HCs and HPs

First, we scanned the expression profiles of all transcripts in CD4 T-cells from two HCs, two ECs, and two HPs. In all, more than 82000 mRNA transcripts and more than 20,000 corresponding genes were identified. Likewise, nearly 15,000 lncRNAs and about 3500 corresponding gene symbols were identified. Differential expression of transcripts between HCs and HPs was first considered. Altogther, 3602 mRNA transcripts and 383 lncRNA transcripts were differently expressed according to the threshold of fold change >2 and corrected $p$ value <0.05 (FDR < 0.05), among which 2313 mRNAs and 207 lncRNAs were upregulated and 1289 mRNAs and 176 lncRNAs were downregulated. Similarly, we

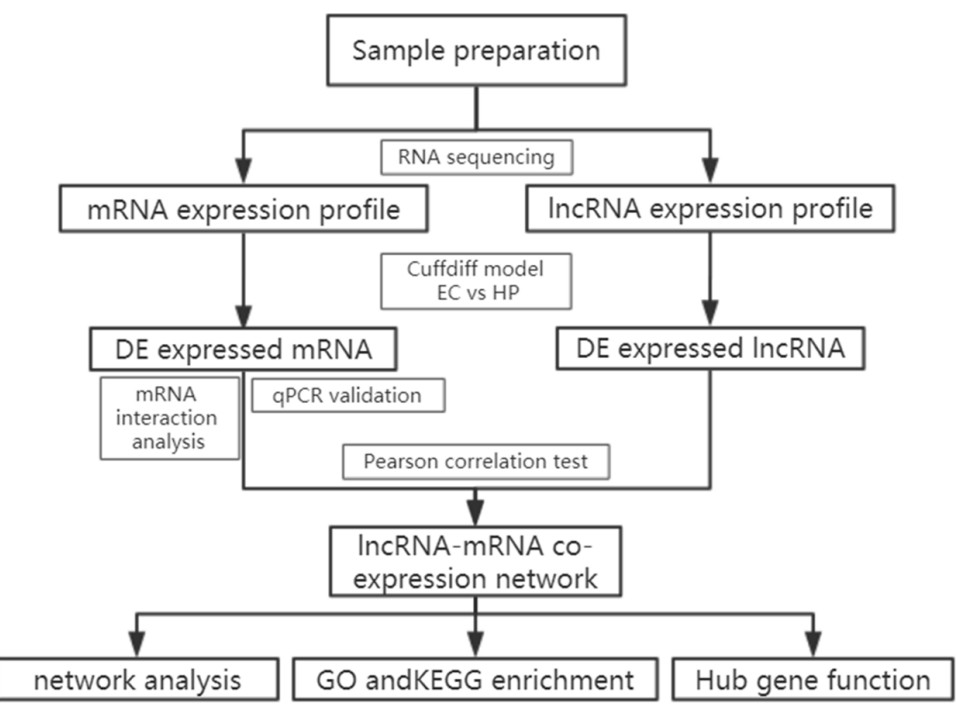

**Figure 1 Flowchart of the analysis procedure.** lncRNA, long non-coding RNA; EC, elite controller; HP, HIV-positive infected patient; DE, differentially expressed; GO, gene ontology; KEGG, Kyoto Encyclopedia of Genes and Genomes.

compared gene expression differences between HCs and ECs, but paid more attention to ECs vs. HPs. The comparison with HCs helped to provide a basic reference background. All transcript FPKM values and summary of gene FPKM values were included in Table S2. The lists of FPKM values and regulatory of all DE mRNAs and DE lncRNAs in HCs vs. HPs and HCs vs. ECs are shown in Tables S3 and S4, respectively.

## Identification of DE mRNAs between ECs and HPs

Among all transcripts, 2936 transcripts were differentially expressed according to the threshold of fold change >2 and corrected $p$ value <0.05 (FDR < 0.05). There were 1327 mRNAs that were upregulated and 1609 that downregulated in ECs vs. HPs. GO and KEGG pathway analysis was carried out to annotate the function of these DE mRNAs. Certain terms and pathways that might be associated with infection and immunity were significantly enriched (Fig. S1). These terms and pathways included negative regulation of interferon-beta production (GO:0032688), cellular component disassembly involved in execution phase of apoptosis (GO:0006921), regulation of defense response to virus by virus (GO:0050690), negative regulation of type I interferon production (GO:0032480), RIG-I-like receptor signaling pathway (hsa04622), and antigen processing and presentation (hsa04612). Note that when considering the overall expression of a gene but encountering

conflicting expression levels of multiple transcripts, we considered the summary of post-merger FPKM values synthetically and prioritized the more statistically significant transcript as the study object.

## Identification of DE lncRNAs between ECs and HPs

Among all transcripts, 3,543 official gene symbols that matched in the NONCODE database had corresponding lncRNA transcripts. Three main categories of lncRNAs accounted for the majority, including antisense lncRNAs (2537/14853, 17.1%), intronic lncRNAs (8152/14853, 54.9%), and long intergenic noncoding RNAs (lincRNAs; 3344/14853, 22.5%). There were 151 upregulated lncRNAs and 160 downregulated lncRNAs in ECs vs. HPs according to the threshold of fold change >2 and $p < 0.05$. In both the mRNA and lcnRNA clusters (Fig. 2), the samples showed marked intra-group correlations and inter-group differences, which further confirmed the specificity of the samples. In terms of the overall profiles of mRNAs encoding proteins with specific biological functions, the HC and EC groups showed closer clustering and were relatively distant from the HP group. The lists of FPKM values and regulatory situations of all DE mRNAs and lncRNAs in ECs vs. HPs are shown in Table S5.

## LncRNA-mRNA co-expression network analysis

Almost 0.9 million co-expression relationships were discovered, and the exact form was expressed as a linear correlation whereby one lncRNA targets one mRNA.Based on previous DE analysis, 1109 DE lncRNAs targeting DE mRNAs were identified In all correlations (Table S6). Figure 3 shows the network constructed using these relationships, in which 142 DE lncRNAs were co-expressed with 127 DE mRNAs. To investigate the importance ranking of these DE mRNAs, and find the genes at the center of the functions of gene expression, we constructed a PPI network. Figure 4 shows the top 50 hub genes that playing an important role in interaction of 127 DE mRNAs, in which many genes functioned centrally and could be frequently regulated by other transcripts. Considering the differences in the outputs of different algorithms, we identified those mRNAs that were commonly detected by all 12 algorithms and focused on those mRNAs that were regulated by lncRNAs and their functional enrichment analysis. Table S7 shows the list of 24 intersecting hub mRNAs among the top 50 calculated using the 12 algorithms. In Fig. 3, we identified 16 of 24 intersecting hub mRNAs in the diagram, which helped us to quickly locate the target protein coding genes and easily access the enriched functions of these genes. Next, we selected the 16 hub mRNAs and their associated lncRNAs from Fig. 3 and constructed Fig. 5. We then identified the top 10 GO terms and pathways of all DE mRNAs most closely related to infection and immunity in Fig. 3. Many significant functions are co-enriched, such as immune response-regulating signaling pathway (GO:0002764) and cellular response to interleukin-1 (GO:0071347) (Fig. 6). It should also be pointed out that when there were various many-to-many regulatory relationships, the most statistically significant regulatory relationship was selected as the main study object.

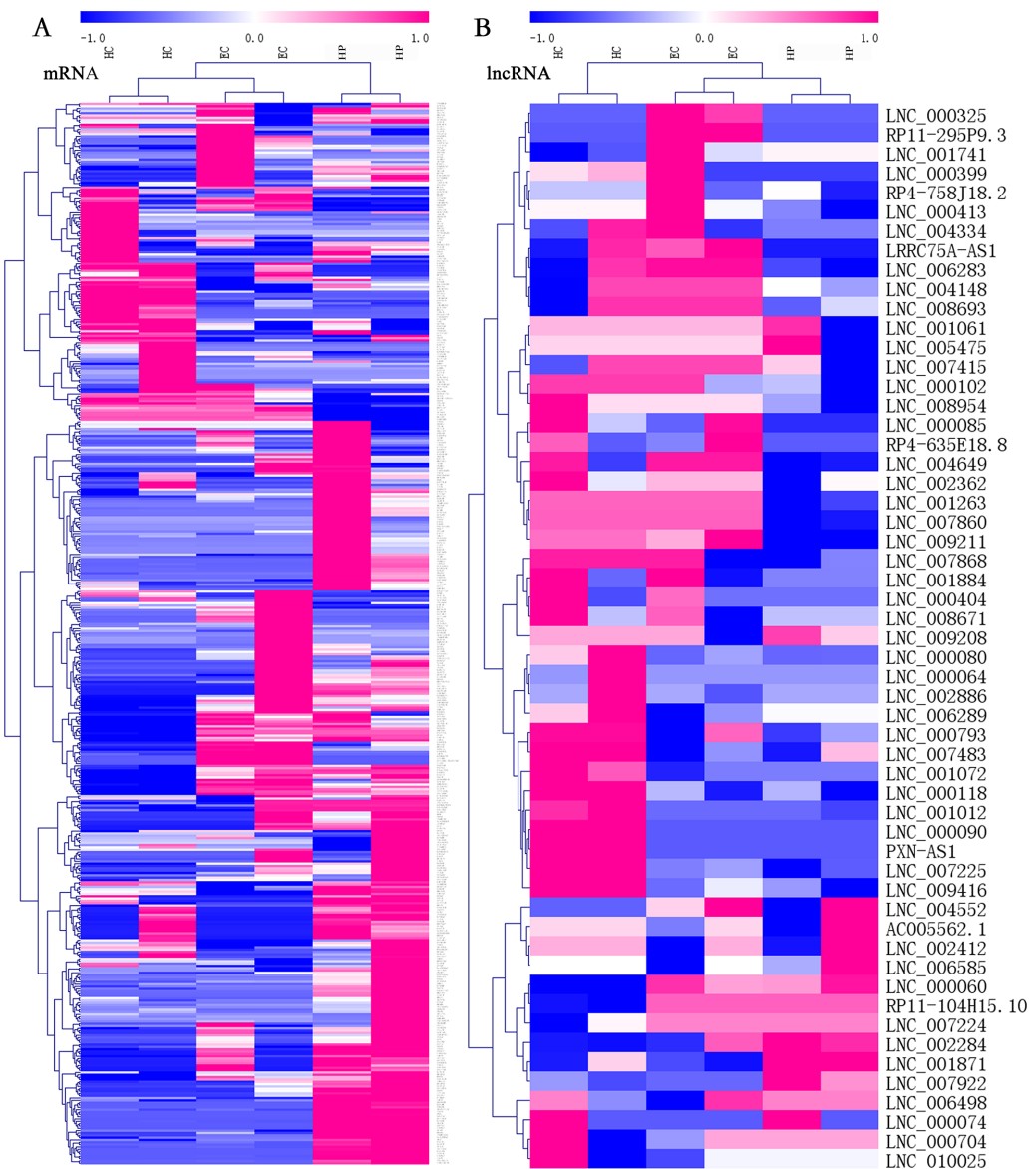

**Figure 2 Heatmaps of differentially expressed (DE) mRNAs and long non-coding RNAs (lncRNAs).** Hierarchical clustering analysis of sequencing detected mRNAs and lncRNAs with their expression abundance. Patient groups are on the horizontal axis, and mRNA and lncRNA genes are grouped along the vertical axis. Pinkish-purple bars indicate increased abundance of the corresponding genes, and blue bars indicates decreased abundance. White bars indicate that the corresponding genes were not detected. (A) Expression abundance of mRNA genes between HC vs. EC vs. HP. (B) Expression abundance of lncRNA genes between HC vs. EC vs. HP.

## Immune regulation and viral infection associated lncRNA-mRNA networks

We further investigated the specific functions of the co-expressed hub mRNAs shown in Fig. 5 and found several interesting regulatory relationships. Certain genes were enriched in pathways that are closely related to immune regulation and viral infection. These genes
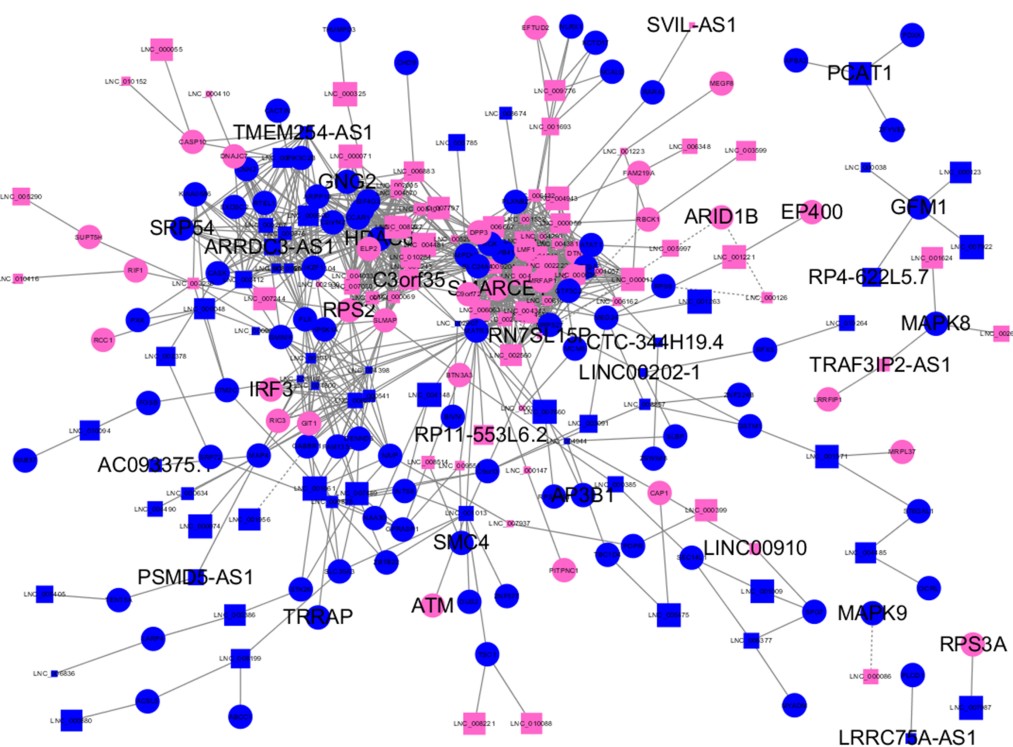

**Figure 3** **Long non-coding RNA (lncRNA)-mRNA co-expression network in elite controllers (ECs) and HIV-positive infected patients (HPs).** The network was based on the Pearson relation coefficient of an lncRNA targeting an mRNA. Round shapes correspond to mRNAs, and squares to lncRNAs. The size of the shape is positively related to the *P* value. Red means upregulation and green means downregulation. A solid line indicates a positive correlation and a dotted line indicates a negative correlation. All hub-mRNAs and annotated lncRNAs are tagged with large labels. All lncRNAs and their co-expressed mRNAs are represented.

and corresponding pathways are shown in Table 1. There were numerous annotations of enriched pathways recurring for different mRNAs, which meaned these hub mRNAs had a close functional relationship. The corresponding annotated lncRNAs, including *C3orf35, TMEM254-AS1, ARRDC3-AS1, LINC00202-1, TRAF3IP2-AS1, RN7SL15P* and *RP4-622L5.7,* were co-expressed with these mRNAs, which suggested several new regulatory relationships that might play an important role in controlling virus replication.

## DICUSSION

In the present study, we systematically analyzed the expression profiles of lncRNAs and mRNAs from CD4+ T cells from ECs, HPs, and HCs, and constructed a co-expression network based on the relationships among DE transcripts and database annotations. The functions of most lncRNAs have not been well characterized. Therefore, the gene functions in this research mainly referred to those of mRNAs and their expression products. The potential functions of candidate lncRNAs and the lncRNAs-associated biological processes in ECs were predicted using the lncRNA-mRNA network and functional enrichment analysis. We identified certain genes that play a significant role in the ability of ECs to

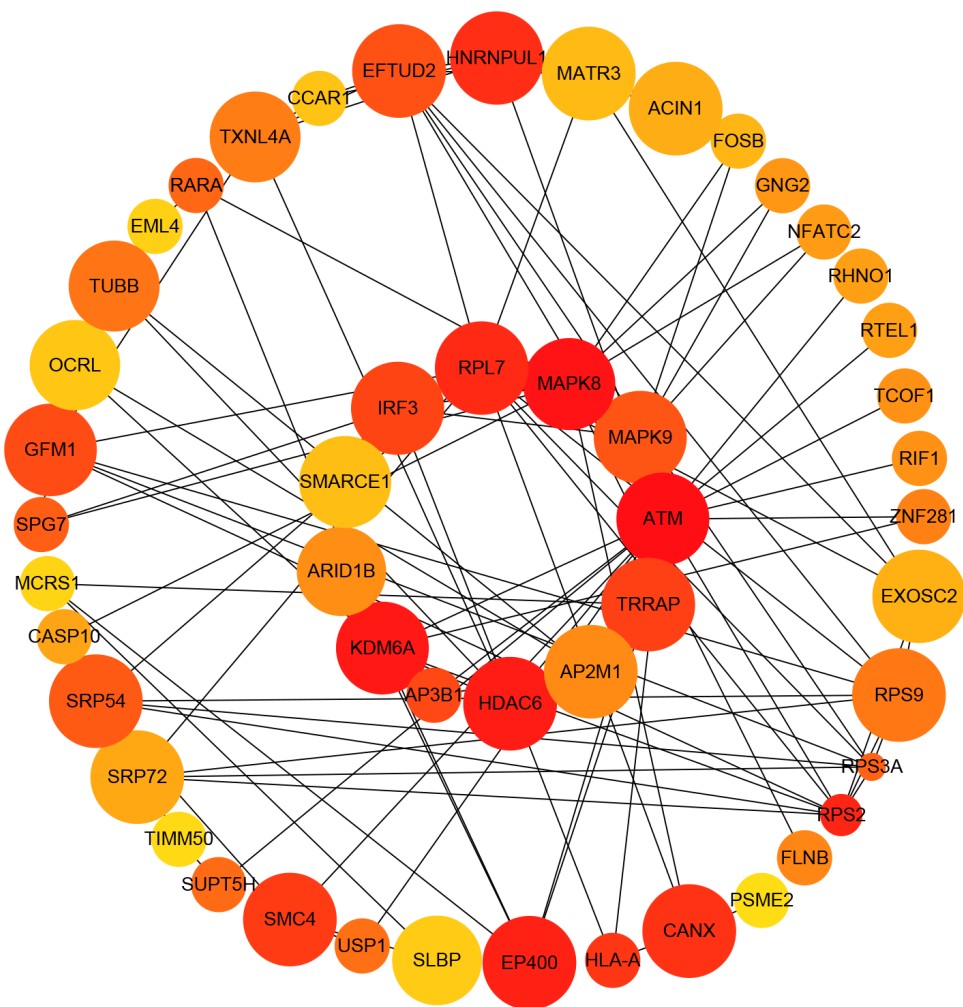

**Figure 4** **Protein–protein interaction (PPI) network of differentially expressed (DE) mRNAs.** The top 50 hub mRNAs were included using the Closeness algorithm. The intensity of the color indicates the density of the correlation. The closer to red, the more central the gene is. The size of the circle represents the significance of a gene.

inhibit the replication of latent virus. Most of these genes were downregulated in the HCs and ECs and upregulated in the HPs or vice versa, which was consistent with their clinical characteristics, in that ECs maintained a similar gene expression pattern to the HCs, which distinguished them from the HPs.

To the best of our knowledge, the present study was the first to predict lncRNA-mRNA interactions for HIV ECs, in which many genes had not been previously reported to function in HIV-associated biological processes. The *MAPK* (mitogen activated protein kinase) family was found to be associated with the reactivation and replication of the virus in many studies (*Wang et al., 2017*; *Prasad & Tyagi, 2015*) , and the HIV-1 virus inhibition activity of *MAPK-NF-κB/AP-1* pathway inhibitors was the result of the negative regulation of HIV-1 LTR promoter activity (*Gong et al., 2011*). In a recent study, *HDAC6* (histone deacetylase
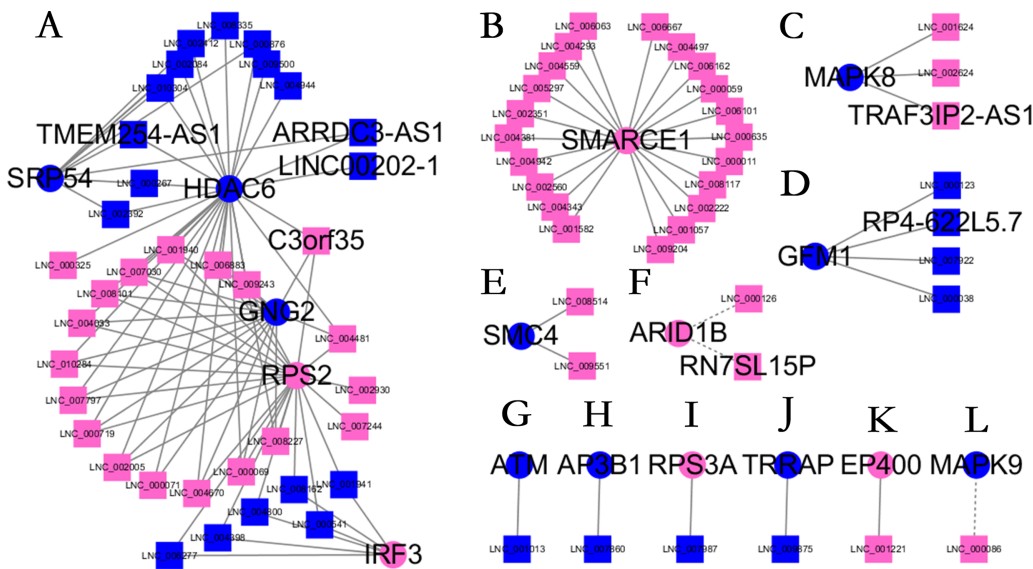

**Figure 5  Long non-coding RNA (lncRNA)-Hub mRNA co-expression network in elite controllers (ECs) and HIV-positive infected patients (HPs).** All hub mRNAs are presented in Fig. 3 together with their associated lncRNAs. (A) *HDAC6, SRP54, GNG2, RPS2, IRF3* and their associated lncRNAs. (B) *SMARCE1* and its associated lncRNAs. (C) *MAPK8* and its associated lncRNAs. (D) *GFM1* and its associated lncRNAs. (E) *SMC4* and its associated lncRNAs. (F) *ARID1B* and its associated lncRNAs. (G) *ATM* and *LNC_001013*. (H) *AP3B1* and *LNC_007860*. (I) *RPS3A* and *LNC_007987*. (J) *TRRAP* and *LNC_009875*. (K) *EP400* and *LNC_001221*. (L) *MAPK9* and *LNC_000086*.

6) overexpression significantly enhanced the expression of pro-inflammatory cytokines, such as TNF-α (tumor necrosis factor alpha), interleukin (IL-1)β, and 289 IL-6, with a concomitant reduction in acetylated α-tubulin. HIV-1 trans-activator of transcription (Tat) can upregulate the expression of various chemokine genes including *IL-1*, *IL-6*, *CCL2* (C-C motif chemokine ligand 2), *CXCL8* (C-X-C motif chemokine ligand 8), and *CXCL10* (C-X-C motif chemokine ligand 10). In addition, *HDAC6* overexpression increased the activation of MAPK factors including extracellular signal-regulated kinase (ERK), Jun terminal kinase (JNK), and mitogen-activated protein kinase 14 (p38). Similarly, *HDAC6* overexpression resulted in activation of the nuclear factor kappa B (NF-κB) and activator protein 1 (AP-1) signaling pathways (*Youn et al., 2016*; *Nookala & Kumar, 2014*). In our results, low expression levels of *HDAC6* and *MAPK8* and downregulation of IL-1 and IL-6 associated receptors in ECs indicated suppressed HADC6- MAPK-NF-κB/AP-1 signaling and fewer immune response disorders. Inhibition of histone deacetylase contributes to reverse the latency of HIV. Ataxia Telangiectasia Mutated (*ATM*) is also involved in the NF-kappa B signaling pathway (hsa04064) by activating p38MAPK and p53 sequentially to induce a pro-apoptotic pathway. *Ariumi & Trono (2006)* indicated that *ATM* could enhance HIV-1 replication by stimulating the action of the Rev viral posttranscriptional regulator, and *Lau et al. (2005)* demonstrated that a novel and specific small molecule inhibitor of ATM kinase activity, KU-55933, is capable of suppressing the replication of

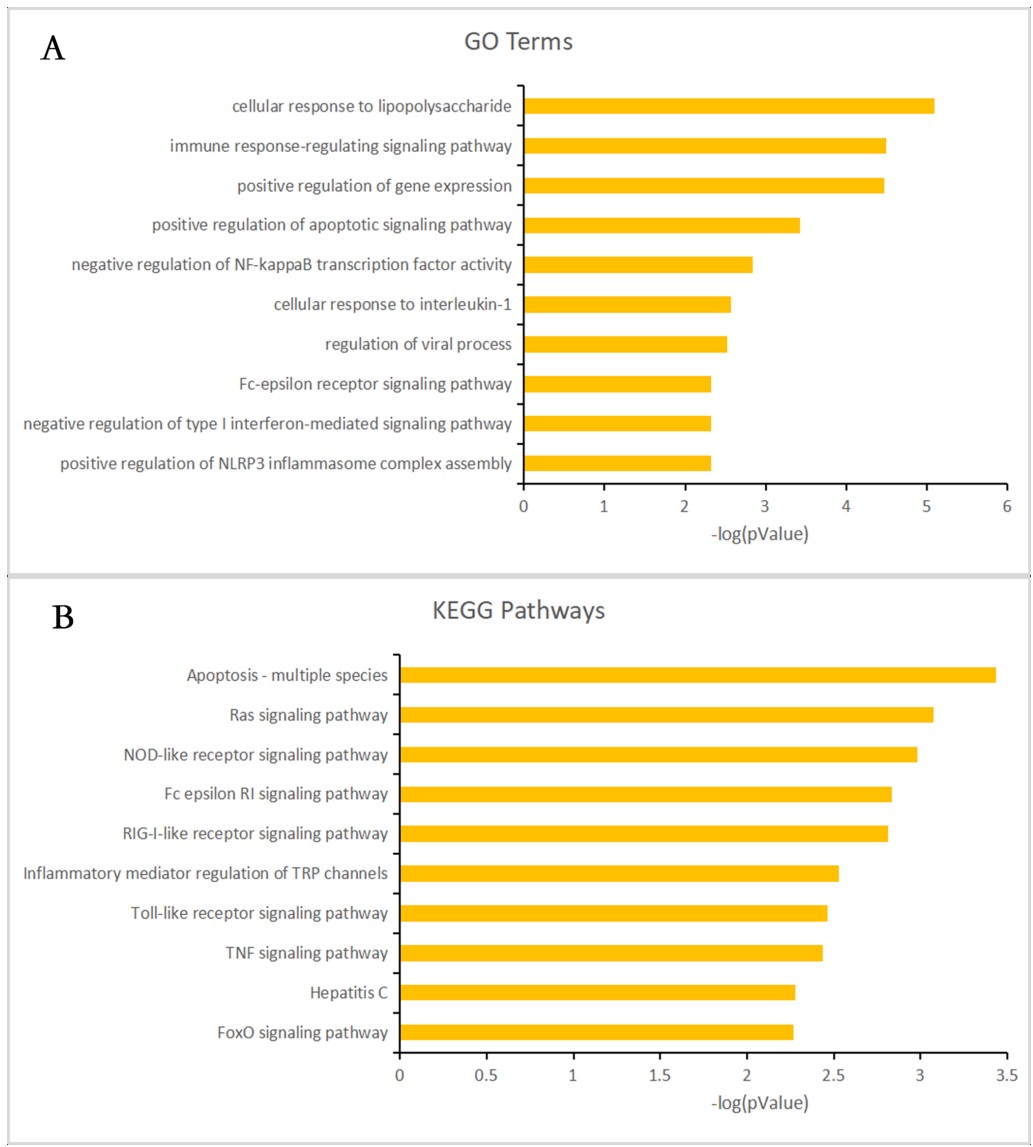

**Figure 6** **Gene ontology (GO) terms and Kyoto Encyclopedia of Genes and Genomes (KEGG) enrichment classification of the predicted targeting mRNAs of the long non-coding RNA (lncRNA)-mRNA network.** The 10 most significantly enriched KEGG pathways and the 10 most significantly enriched GO terms in the biological process, cellular components, and molecular function categories. (A) The top 10 GO terms enriched for predicted targeting mRNAs of the lncRNA-mRNA network. (B) The top 10 KEGG terms enriched for predicted targeting mRNAs of the lncRNA-mRNA network.

both wild-type and drug-resistant HIV-1. Interestingly, some studies considered that HIV-encoded protein Vpu contributes to the attenuation of the anti-viral response by partial inactivation of interferon regulatory factor 3 (IRF3) while host cells undergo apoptosis (*Park et al., 2014*). By contrast, another study reported that Vpu affects interferon expression by inhibiting NF-κB activity without affecting IRF3 levels or activity (*Manganaro et al., 2015*). Thus, the role of IRF3 in HIV infection and replication remains controversial. The

**Table 1  Immune regulation and viral infection associated lncRNA-mRNA networks.**

| Hub Gene | Function Annotation | KEGG Entry |
|---|---|---|
| GNG2 | Human cytomegalovirus infection | hsa05163 |
|  | Chemokine signaling pathway | hsa04062 |
|  | Kaposi sarcoma-associated herpesvirus infection | hsa05167 |
|  | Human immunodeficiency virus 1 infection | hsa05170 |
| ATM | NF-kappa B signaling pathway | hsa04064 |
|  | FoxO signaling pathway | hsa04068 |
|  | Apoptosis | hsa04612 |
|  | Human T-cell leukemia virus 1 infection | hsa05166 |
|  | Human immunodeficiency virus 1 infection | hsa05170 |
| HDAC6 | Viral carcinogenesis | hsa05203 |
| MAPK8 | Apoptosis - multiple species | hsa04215 |
|  | NOD-like receptor signaling pathway | hsa04621 |
|  | Fc epsilon RI signaling pathway | hsa04664 |
|  | RIG-I-like receptor signaling pathway | hsa04622 |
|  | Inflammatory mediator regulation of TRP channels | hsa04750 |
|  | Toll-like receptor signaling pathway | hsa04620 |
| SMC4 | Epstein-Barr virus infection | hsa05169 |
| TRRAP | Human T-cell leukemia virus 1 infection | hsa05166 |
| IRF3 | RIG-I-like receptor signaling pathway | hsa04622 |
|  | Herpes simplex infection | hsa05168 |
|  | Human papillomavirus infection | hsa05165 |
|  | Influenza A | hsa05164 |
|  | NOD-like receptor signaling pathway | hsa04621 |
|  | Toll-like receptor signaling pathway | hsa04620 |
| MAPK9 | TNF signaling pathway | hsa04668 |
|  | Focal adhesion | hsa04510 |
|  | IL-17 signaling pathway | hsa04657 |
|  | Th1 and Th2 cell differentiation | hsa04658 |
|  | Th17 cell differentiation | hsa04659 |
|  | T cell receptor signaling pathway | hsa04660 |

above hub mRNAs are related to annotated lncRNAs including *C3orf35*, *TMEM254-AS1*, *ARRDC3-AS1*, *LINC00202-1*, *TRAF3IP2-AS1* and to unannotated lncRNAs (Fig. 5), which might control HIV replication by regulating the expression of these genes. To date, no study has been published showing that these lncRNAs function in the pathways identified in the present study.

Alternatively, some hub genes (Fig. 4) were not regulated by lncRNAs but interacted with each other and participated in biological processes such as HIV life circle, inflammation and immune activation and cytokines. Lysine demethylase 6A (KDM6A; also called UTX-1) enzymatic activity is required for the viral protein Tat to remove a repressive mark H3K27me3 in the HIV-1 long terminal repeat (LTR) and promoted HIV-1 gene expression by enhancing both NF-κB p65′s nuclear translocation and its binding to the

HIV-1 LTR. H3K27 demethylase activity is required for increased HIV-1 transactivation induced by *KDM6A* (*Zhang et al., 2016*). In our study, the low expression level of *KDM6A* in the ECs might be a possible causes of their low viral loads. *TRAF3IP2* encodes TRAF3 interacting protein 2, wich is involved in regulating responses to cytokines by members of the Rel/NF-kappaB transcription factor family. TRAF3IP2 interacts with TRAF proteins (tumor necrosis factor receptor-associated factors) and either I-kappaB kinase or MAP kinase to activate either NF-kappaB or Jun kinase. We found that lncRNA *TRAF3IP2-AS1* was an anti-sense non-coding RNA of *TRAF3IP2* that was upregulated in the ECs, and was co-expressed with downregulated *MAPK8*. This suggested there may be a regulatory relationship among *TRAF3IP2-AS1*-TRAF3IP2-MAPK8 that plays an important role in the suppression of viral replication.

HAART cannot eradicate the latent virus in the host; therefore, the genes identified in the present study might play a non-negligible regulatory role in the relationship between the latent virus and the host. Latent virus is the biggest obstacle to the elimination of the disease. Thus, understanding the processes that contribute to its persistence, such as inflammation and immune activation, are crucial for the remission and cure of HIV (*Dahabieh, Battivelli & Verdin, 2015*; *Massanella, Fromentin & Chomont, 2015*) .

There are also some limitations to this study. First, because of the limited amount of RNA in the samples, the sample size of this study was relatively small. Although we selected eight and six mRNAs for verification using qRT-PCR for 16 genes and 40 samples three times, respectively and found that the expression change trends of 13 of them were consistent with the RNA sequencing results, these results lack of sufficient statistical significance because of the small sample size. Part of the verification analysis is shown in Fig. S2. Second, we only assessed gene expression of CD4+ T cells, which, although they are the main targets of HIV and are vitally important, they are only one of the many cell types that are infected by HIV. Further research should be performed to validate the function and mechanism of the genes in a larger sample and in more cell types.

## CONCLUSIONS

This study identified several important differentially expressed genes associated with the elite controller phenomenon, using RNA-sequencing and bioinformatics analysis to explore how ECs spontaneously control virus replication. Some differentially expressed genes were identified and enriched for meaningful GO terms and KEGG pathways related with viral infection and immune responses. After filtering lncRNAs and co-expressed mRNAs using the Pearson correlation test, the functions of the lncRNAs were predicted using an lncRNA-mRNA network comprising the functional annotations of mRNAs. This study forms the basis for subsequent cellular and molecular studies, and provides new targets for gene-targeted therapy in the future.

# ACKNOWLEDGEMENTS

We would like to thank all of the participants in the cohort. In addition, we would like to thank the native English speaking scientists of Elixigen Company (Huntington Beach, California) for editing our manuscript.

## Funding

This study was supported by grants from the Mega-Project for National Science and Technology Development under the "Study on comprehensive prevention and treatment of AIDS in demonstration areas, 13th, Five-Year Plan of China" (NO.2017ZX10105001-005) and "Research and application of appropriate treatment and prevention strategies for children with AIDS, 13th, Five-Year Plan of China" (NO.2018ZX10302-102). There was no additional external funding received for this study. The funders had no role in study design, data collection and analysis, decision to publish, or preparation of the manuscript.

## Grant Disclosures

The following grant information was disclosed by the authors:
Mega-Project for National Science and Technology Development: 2017ZX10105001-005.
Research and application of appropriate treatment and prevention strategies for children with AIDS, 13th, Five-Year Plan of China: 2018ZX10302-102.

## Competing Interests

The authors declare there are no competing interests.

## Author Contributions

- Chaoyu Chen conceived and designed the experiments, performed the experiments, analyzed the data, prepared figures and/or tables, and approved the final draft.
- Xiangyun Lu performed the experiments, analyzed the data, prepared figures and/or tables, and approved the final draft.
- Nanping Wu conceived and designed the experiments, authored or reviewed drafts of the paper, and approved the final draft.

## Human Ethics

The following information was supplied relating to ethical approvals (i.e., approving body and any reference numbers):

Ethical approval was granted by the Ethics Committee of the First Affiliated Hospital, College of Medicine, Zhejiang University. The reference number is 2017-338.

## Data Availability

All raw data is available in the SRA at NCBI: PRJNA575767.

## Supplemental Information

Supplemental information for this article can be found online at http://dx.doi.org/10.7717/peerj.8911#supplemental-information.

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
