# Peer review of "RNA sequencing of CD4 T-cells reveals the relationships between lncRNA-mRNA co-expression in elite controller vs. HIV-positive infected patients"

_PeerJ, doi:10.7717/peerj.8911_

## Round 0.1 · original submission · Major Revisions

Three specialists in the field evaluated this submission. They all have concerns related to this manuscript. Please ensure that the English language in this submission meets our standards: uses clear and unambiguous text, is grammatically correct, and conforms to professional standards of courtesy and expression. Considering the evaluation carried out by these reviewers, I recommend major revision in this submission.

Reviewer 1 ·

Basic reporting

The author described the relationships between lncRNA-mRNA co-expression in elite controller vs. normal-process HIV patients. They performed bioinformatics and RNA-seq to identify differentially expressed mRNAs and lncRNAs from CD4+ T cells from ECs, NPs, and HCs, and construct a PPI network. Finally, they obtained some hub genes associated with viral infections and immune responses. However,the manuscript had some writing and logic mistakes and should be revised (e.g. the rather than th).

Experimental design

Why authors just collected CD4 T-cell samples from two ECs, two NPs, and two HCs to perform second-generation transcriptome sequencing? I think the sample size is too small to be statistical significance.

Validity of the findings

The article focused on bioinformatics and was lack of in vivo and in vitro experiments to confirm the final conclusions.

Reviewer 2 ·

Basic reporting

This manuscript by Chen et al titled 'RNA sequencing of CD4+ T cell reveals the relationshis between lncRNA-mRNA co-expression in elite controller vs normal-rocess HIV patients' reports differential expression of lncRNAs and protein-coding mRNAs in CD4+ T cells from elite controllers, HIV-patients (NP) and Healthy controls (HC). Two individuals, one male and one female are included in each group.
Major concern:
1. Given the inter-individual variability in gene expression, the small number of samples are insufficient for such analyses.
2. Just one male and female in each group are not sufficient account for gender-based differences in transcriptome.

Experimental design

Major concerns
1. Authors describe 'corrected p value' but the correction is not explained in the methodology. it is unclear if the corrected p value takes false discovery rate (FDR) into consideration.
2. it is unclear if

Validity of the findings

1. In the raw data files, please mention genomic co-ordinates of potentially novel lncRNAs for which there are no gene or transcript identifiers
2.A gene may make multiple transcript isoforms. If some of the transcript isoforms are differentially expressed as in "Differentially_Expressed_Transcripts", the sum of all the transcript isoforms for that gene may not be deferentially expressed (statistically significant). Authors seem to have considered only the statistically significant transcript
(line 211-213). This is unconventional and inaccurate unless the expression or function of the gene is driven by the differentially expressed isoform.

Additional comments

Minor
grammatical and typographical errors in the manuscript

Reviewer 3 ·

Basic reporting

The manuscript needs careful editing for English language (including many spelling mistakes), but also for correct use of terminology (e.g. what is normal-process HIV patient?) and consistency in terminology (for the same therapy the authors use anti-viral therapy, antiretroviral therapy (ART) and highly active antiretroviral therapy (HAART)). Please also focus on improving the title and abstract. The results in paragraph under "Immune regulation and viral infection associated lncRNA-mRNA networks" should be moved/written in a form of the Table for better clarity. The Discussion section is too long.

Experimental design

The authors investigate the relationship between lncRNA and mRNA in CD4 T cells from 6 donors (two per each patient group) to better understand the natural ability of elite controllers to repress HIV replication.
As HIV infected individuals from the same patient group can vary greatly in their clinical characteristics (also holds true for elite controllers as they likely employ different mechanisms for repression of HIV replication), sequencing transcriptome of two patients per group is not enough to make any relevant comparisons and statements. The authors should include more patients per group OR they should validate their sequencing/analysis outcomes on a larger group of patients by qPCR for few target lncRNA/genes. As 196 patients were included in the study that shouldn't be a problem.
The authors should also better explain the differences between patients under treatment and untreated normal process patients. Are normal process patients infected but untreated patients with high viral load? If so, it would make much more sense to compare ART treated infected individuals vs. elite controllers to make any conclusions on how elite controllers repress HIV replication.
The authors should also include in the Method section how the data was normalized to enable comparisons between patients and between groups.
Please add and discuss the paper on lncRNA HEAL regulation of HIV replication (PMID: 31551335).

Validity of the findings

No further comment beside the comment under Experimental design.

---

## Round 0.2 · Major Revisions

One of the reviewers said that the discussion section is still too long and not closely related to the topic. Please revise the manuscript.

Reviewer 1 ·

Basic reporting

The discussion part is still too long and not closely related to the topic. The author should pay attention to highlight the key points.

Experimental design

Although the author replied that biological repeats did not seem to be necessary in some earlier studies, the examples are research about plants. As is known, plants are more biologically conservative than animals, so I think you should also expand the sample size. Considering the limited sample size, maybe you can use bioinformatics to mine published data.

Validity of the findings

no comment

Additional comments

The author described the relationships of lncRNA-mRNA in elite controller vs. HIV-positive infected patients. They performed bioinformatics and RNA-seq to identify differentially expressed mRNAs and lncRNAs from CD4+ T cells from ECs, HPs, and HCs, and construct a PPI network. Finally, they obtained some hub genes associated with viral infections and immune responses. However, there are still some parts to be revised.

---

## Round 0.3 · accepted · Accept

The authors carried out all modifications requested by the referees in all stages of the evaluation of this submission. In my view, the manuscript improved during the revision process. It can be accepted as it is.